# Impacts of Shade Trees on the Adjacent Cacao Rhizosphere in a Young Diversified Agroforestry System

Jennifer E. Schmidt [1,*] , Alana Firl [1], Hamran Hamran [2], Nur Insana Imaniar [2], Taylor M. Crow [1] and Samantha J. Forbes [3]

1   Cocoa Plant Sciences, Mars Wrigley, Davis, CA 95616, USA; alana.firl@effem.com (A.F.);
    taylor.crow@effem.com (T.M.C.)
2   Mars Indonesia, KIMA X Kav A6, Makassar 90241, Indonesia; hamran.hamran@effem.com (H.H.);
    nur.imaniar@effem.com (N.I.I.)
3   Mars Sustainable Solutions, Cairns 4878, Australia; samantha.forbes@effem.com
*   Correspondence: jennifer.schmidt@effem.com

**Abstract:** Cacao agroforestry systems offer the potential to diversify farmer income sources, enhance biodiversity, sequester carbon, and deliver other important ecosystem services. To date, however, studies have emphasized field- and system-scale outcomes of shade tree integration, and potential impacts on the rhizosphere of adjacent cacao trees have not been fully characterized. Interactions at the root–soil interface are closely linked to plant health and productivity, making it important to understand how diverse shade tree species may affect soil fertility and microbial communities in the cacao rhizosphere. We assessed the impacts of neighboring shade tree presence and identity on cacao yields and physical, chemical, and biological components of the cacao rhizosphere in a recently established diversified agroforestry system in South Sulawesi, Indonesia. Stepwise regression revealed surprising and strong impacts of microbial diversity and community composition on cacao yields and pod infection rates. The presence of neighboring shade trees increased nitrogen, phosphorus, and pH in the rhizosphere of nearby cacao trees without yield losses. Over a longer time horizon, these increases in rhizosphere soil fertility will likely increase cacao productivity and shape microbial communities, as regression models showed nitrogen and phosphorus in particular to be important predictors of cacao yields and microbiome diversity and composition. However, neither presence nor identity of shade trees directly affected microbial diversity, community composition, or field-scale distance-decay relationships at this early stage of establishment. These results highlight locally specific benefits of shade trees in this agroecological context and emphasize the rhizosphere as a key link in indirect impacts of shade trees on cacao health and productivity in diversified systems.

**Keywords:** agroforestry; cacao (*Theobroma cacao*); diversification; microbiome; rhizosphere; shade trees; soil health

## 1. Introduction

Cacao (*Theobroma cacao*, L.) agroforestry systems (CAFS), in which cacao grows under one or more tree species, are a biodiverse alternative to monoculture production. Integration of shade trees provides ecosystem services such as carbon sequestration, preservation of biodiversity, and pest management, although in some cases these may come at the expense of short-term productivity [1–3]. Shade trees benefit cacao by favorably modifying the microclimate: buffering temperature extremes and wind, decreasing erosion locally, and reducing incoming light to avoid unwanted vegetative growth [4].

Positive impacts of shade trees on soil fertility are frequently cited as a benefit of CAFS, as leaf litter and decomposing fine roots generate an important influx of nutrients into the soil. Inputs of C from leaf litter, fine roots, and other decomposing mass can also be stored in soil aggregates and contribute to C sequestration [5]. However, decomposition, nutrient mineralization rates, and impacts on soil fertility and functions can vary by shade

species or combination of species [6–9]. This variation may be due to qualitative and quantitative differences in leaf litter [10], including the C:N ratio, C:P ratio, concentrations of polyphenols, and presence of other nutrients that may be limiting [6].

Diversification of tree species in CAFS impacts not only soil physicochemical properties, but also living components of the agroecosystem. Shade trees can impact biodiversity and composition of much smaller biota even despite orders-of-magnitude higher species richness in the latter community [10]. Presence and identity of shade trees affects earthworm and soil microarthropod community structure in some CAFS [10,11], though soil macroinvertebrates are not always affected by shade species identity [12]. In contrast, shade tree species diversity does not affect microbial community phospholipid fatty acid (PLFA) profiles or microbial biomass across a diversity gradient in Southeast Sulawesi [13].

Impacts of shade trees in CAFS have been measured primarily at the tree scale (under shade tree canopies) [9,14], or at the plot scale in comparisons among CAFS [8,15] or between CAFS and monoculture [2]. Impacts of shade trees on soil physicochemical properties at the tree scale may not be observed at the plot scale, however [16], highlighting a need for further research across scales. Furthermore, studies to date have not investigated how shade tree integration may affect the rhizosphere of adjacent cacao trees. Given that cacao productivity is a key outcome of interest to ensure the economic livelihood of millions of cacao farmers, and that the rhizosphere is a zone with disproportionate influence on plant health and productivity in comparison to bulk soil, this may be a knowledge gap of great economic and ecological relevance.

The present study sought to (1) understand how presence and identity of diverse shade trees impacts soil physical, chemical, and microbial characteristics in the adjacent cacao rhizosphere; and (2) characterize direct and indirect relationships among shade trees, soil abiotic properties, microbial communities, and cacao yields. We hypothesized that shade tree presence and species identity would directly impact cacao yields (Figure 1). Furthermore, we hypothesized that variation among shade tree species would affect cacao rhizosphere properties relevant to yields, including soil fertility and microbial diversity, community composition, and spatial distribution within the plot (Figure 1).

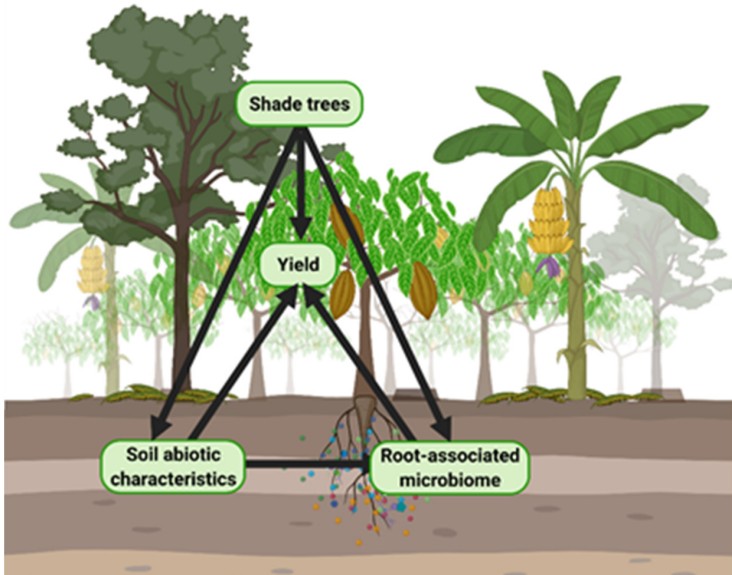

**Figure 1.** Hypothesized relationships of shade trees, soil physicochemical properties, root-associated microbial communities, and yields in cacao agroforestry systems. Shade trees are predicted to directly impact yields by altering microclimatic conditions and available photosynthetically active radiation, and impact soil properties and microbial communities via litter inputs and rhizosphere effects. Both yields and microbial communities are expected to respond to nutrient availability and other soil abiotic characteristics. Beneficial and pathogenic microorganisms are expected to increase and decrease cacao yields, respectively.

## 2. Materials and Methods

### 2.1. Site Location and Environment

The study was carried out in a highly diversified cacao plot in South Sulawesi, Indonesia. Indonesia has experienced a decline in cacao production area and volume in recent years after ranking third among cacao-producing countries from 1995 to 2015 [17]. Ninety percent of Indonesian cacao production occurs on smallholder farms [18], where shade trees are frequently planted at least in the initial phase of cacao establishment [19].

The diversified study plot was established in 2017 on a recently cleared (~3 years noncultivated) 0.5 Ha plot within the Mars Cocoa Research Centre (MCRC), in Tarengge Village, South Sulawesi, Indonesia (2°33′09″ S, 120°48′21″ E). The MCRC has an average annual rainfall of 2955.2 mm, well-distributed over 12 months, with higher than average monthly rainfall between April and June (average 421.3 mm) and lower than average rainfall between August and November (average 94.7 mm). The annual average temperature ranges between a minimum of 19.9 °C and a maximum of 36.4 °C, annual relative humidity is between 43.3 and 100%, and the average monthly solar radiation is 217 W/m$^2$. During the year of sampling for the current study, the average annual rainfall was 311.22 mm (wet) and 101.26 mm (dry), and 27.46 °C (wet) and 26.24 °C (dry), respectively. The soils at the MCRC are generally classified as Typic Dystrudepts in the USDA system of soil taxonomy, very homogeneous and with little spatial variation.

### 2.2. Site History, Current Trials, and Plot Management

The study plot was cleared with a tractor in 2013. This included complete uprooting and removal of existing trees and their root systems, after 40+ years of mixed durian (*Durio sp*. L.) and coconut (*Cocos nucifera* L.) cultivation, and with naturally occurring recruits of local banana (*Musa sp*. L.) plants and various other non-cocoa tree species (e.g., teak, *Tectona grandis* L.f.). The cultivation of durian and coconut in this region does not involve the application of agrochemicals. Prior to the cultivation of durian and coconut, the diversified plot was used for rice cultivation, with unknown, but likely intensive, agrochemical use. Cassava was cultivated on the site after clearing in 2013, and prior to the establishment of the current trial in 2017. The site is 100 m × 50 m and is oriented N-S on its longitudinal axis.

The initial plot design included 703 plants of 14 different plant species (cacao (*n* = 253), shade trees, fruits, timber, and spices) (a full description of species common names, scientific names, and numbers of plants can be found in Table S1). The arrangement of the selected plant species within the plot was determined by considering (1) species-specific physical and physiological characteristics (e.g., mature height, canopy and root architecture, growth rates, and resource requirements), (2) structural complexity (i.e., do the species provide multi-strata vegetative complexity?), and (3) inter-species interactions (e.g., potential complementarity and mutually beneficial crop interactions, as well as allelopathy and competition). Finally, (4) labor and harvest efficiency were large considerations in the spatial arrangement of differing plant groups and species (e.g., timber trees on plot edges for harvest efficiency and to limit damage upon felling, vegetables close to the rainwater tank for easy watering during the dry season). The cacao in the diversified plot was clonally propagated and included two clones (MCC02, *n* = 244, and BB01, *n* = 9). Scion material from both clones was top grafted onto seed-grown rootstocks from clone BB01 fruits. In addition to these plants, gliricidia sticks were planted at 9 m and 9 m spacing to provide additional (temporary) shade for the developing cacao seedlings. As the permanent shade tree species grew and matured (~2–3 years), the gliricidia plants were removed from the plot to prevent overshading. In preparation of planting (September 2017), the plot soil was turned with a rotary hoe before 2 tons each of dolomite and cow manure were spread evenly across the cleared site and again turned into the soil. This process was conducted to increase the existing soil pH level, which was assumed to be low due to the cultivation history. The planting of all plant species was carried out during a one-month period between November and December 2017. A complete list of soil amendments that were added upon planting for

cocoa, fruit, and timber species is provided in Table S2. No soil amendments were added upon planting of gliricidia sticks.

Since the trial's establishment, the cacao, shade, fruit, and timber trees have been given fertilizers and compost on a regular basis to support tree growth and nutrition (Table S3). The centrally located vegetable plot was managed using regenerative agricultural principles, and the additional annual plant species on the plot borders such as timber, pineapple, chili, and papaya were also managed with fertilizers and pesticides. However, no soil samples were taken from these areas. Mulch (banana leaves and stems, cacao prunings, gliricidia prunings, timber prunings) was applied underneath cacao trees as these species are pruned. Pruning of cacao trees occurred twice annually with one heavy prune (~July) and one light sanitary prune (~December). All prunings were maintained on the plot as mulch on top of the soil surface and prioritizing soil areas above the cacao root zone (underneath the canopy area).

For pest and disease management, Prevathon (DuPont™ Prevathon™, active ingredient Chlorantraniliprole) was used as an insecticide to protect the cacao trees primarily from cocoa pod borer (CPB) (*Conopomorpha cramerella*, Snellen), a major insect pest of cacao, and Score (Syngenta®, active ingredient Difenoconazole) was used as a fungicide to protect cacao trees against Phytophthora (*Phytophthora palmivora*) infection. To apply Prevathon and Score, a 15 L knapsack sprayer was used containing a solution of 1 mL Prevathon and 1.66 mL Score for every 1 L of water. The canopies of cacao trees were sprayed every 2 weeks during the period of growth when cacao pods were most vulnerable to CPB attack (when cacao cherelles reached 1 inch and until harvest). The Prevathon/Score solution was applied as a light mist to the whole tree canopy and directly onto cacao pods.

## 2.3. Soil Sampling and Analysis

Soil samples were collected from around 80 cacao trees of the MCC02 clone, with each tree representing one biological replicate (*n* = 80). Soil physicochemical parameters were analyzed on a subset of these (*n* = 48) due to resource constraints. The location of sampling points followed a semi-stratified design in attempt to capture site variability. Samples were classified as belonging to either of two groups: Diverse-Cacao for cacao trees with adjacent non-cacao species, and Cacao-Cacao for cacao trees surrounded only by other cacao trees. Trees in the Diverse-Cacao group were further classified by the nearest non-cacao species.

For each identified cacao tree, three soil core samples were taken 1 m away from the tree trunk. We used a soil corer $1\frac{1}{4}$ inch in diameter to sample a vertical soil core of 0–10 cm depth. Each soil core was composed of at least 200 g wet soil mass, excluding large stones and rocks. This was to ensure a final total sample mass of 300 g (dry mass) after sample processing and preparation for the composite of three subsamples. The litter layer of organic matter as well as live plants such as grasses covering the soil surface were removed prior to soil sampling. To eliminate contamination between samples, a fresh pair of latex gloves was used for each tree's soil sampling, and the soil sampling tools were thoroughly cleaned (water wash and scrub) and sterilized (ethanol dip and paper towel dry) before each tree's sample was taken, ensuring all tools were completely free from soil and were dry before their next use. The three soil cores per tree were placed together in a Ziplock bag (appropriately labelled) where they were homogenized thoroughly to form one well-mixed representative sample and sealed tightly to ensure that no water loss occurred.

## 2.4. Microbiome Sampling and Sequencing

For the microbial analysis, two (duplicate) subsamples were then extracted from the homogenized sample using fresh gloves, a clean pipette and sterile 2.0 mL tubes and filled immediately with Zymo DNA/RNA Shield buffer (Zymo Research Corporation, Irvine, CA, USA) to completely immerse the ~25 g soil subsample. Tubes were then sealed tightly, shaken vigorously to ensure all soil came into contact with the buffer, and stored in a Styrofoam cooler box in the shade (for no longer than 1 h) and transported to an air-conditioned room until shipping to the lab. Soil samples for microbial analysis were

shipped immediately. The buffer stabilizes the DNA for two weeks at room temperature and the DNA extractions were conducted within two weeks of sample collection. Care was taken to ensure the pipette tip sterile tubes did not come into physical contact with the tube or the soil. Pipette tips and gloves were replaced after every sample and if they came into contact with anything foreign.

DNA was extracted from 0.25 g of soil per subsample using DNeasy Powersoil Pro kits (QIAGEN, Germantown, MD, USA) according to manufacturer's instructions. Sequencing was conducted using 250 bp paired end reads with separate sequencing runs for bacteria and fungi. The original 515F/806R primer pair was used to amplify the V4 region [20] on an Illumina MiSeq platform, and the ITS1F/ITS2 primer pair was used for the ITS1 region on an Illumina Novaseq platform [21,22].

### 2.5. Soil Physicochemical Sampling and Analysis

Immediately after the microbial subsamples were removed, the Ziplock bag containing the homogenized composite soil sample was weighed for wet mass in the field with a transportable scale to $\pm 0.001$ g accuracy. The samples ($n = 48$) were then placed in a cool box out of direct sunlight before they were transported to the lab for further processing. In the lab, composite soil samples were dried in solar dryers for 5 days, or until dry to touch, and "finished off" in drying ovens at 40 °C for 24 h. Samples which could not be directly put into the oven after solar drying were placed in $-20$ °C freezer until there was room in the oven. Once dry, the samples were weighed again for dry mass determination as soon as they were removed from the drying oven. Samples were then double bagged in plastic Ziplock bags and stored in a dark place in an air-conditioned lab until shipping to the Indonesian Coffee and Cocoa Research Institute (ICCRI) laboratory for physicochemical analysis.

Soil texture was characterized by removing the sand fraction with a 50-micron sieve, then measuring silt and clay fractions using a hydrometer. Soil pH was measured in aqueous solution with a combination glass electrode. $NH_4$-N was determined via distillation with boric acid and titration with sulfuric acid, and $NO_3$-N was determined in the remaining extract by reduction with Devarda's alloy and the same titration procedure. Cation exchange capacity (CEC) was measured with the ammonium acetate method. Phosphorus was measured using a spectrophotometer at wavelength 889 nm following extraction of soil with dilute HCl. Potassium and other cations were measured similarly.

### 2.6. Yield Measurements

Cocoa yield was evaluated as the total number of cocoa pods (both healthy and pest- and disease-infected pods) and the dry mass (g) of beans obtained per tree for the main harvest period in 2020. The cocoa trees were only 4 years old and not yet at peak production. In the field, cocoa harvest was conducted on a fortnightly basis. On each harvesting date during the main harvesting period (May–July), any pods that were ripe were harvested, the number of harvested pods was counted, and the pods were categorized as "healthy" OR "infected" by visually inspecting the pods for signs of infections. The pods were then opened on site, and the wet mass (g) of cocoa beans was determined and transformed into dry bean mass (g), using the standard 35% conversion rate as is recognized for clone MCC02 (dry bean mass is typically 34–37% of the original wet bean mass for this clone).

### 2.7. Bioinformatics and Statistical Analysis

Twelve samples did not yield bacterial sequencing data of sufficient quality for use, and one did not yield fungal sequencing data of sufficient quality, so these were not included in the analysis of the remaining samples ($n = 68$ for bacteria, $n = 79$ for fungi). Primers and adapters were removed from raw sequencing reads and low-quality reads (length < 60 bp, Q score < 5, or unidentified bases >10%) were filtered out using fastp software [23]. Subsequent analyses were done in R software v4.0.3 [24], with the dada2 package v1.18 [25] used to filter reads for a maximum of two expected errors, denoise, merge paired-end reads, and remove chimeras. Sequences outside the expected length of

250–256 bp were removed from the 16S dataset, but no length-based filtering was done for the ITS1 sequences because the region varied in length. Amplicon sequence variant (ASV) tables were constructed separately for 16S and ITS1 sequences, and chloroplasts, mitochondria, and archaea were removed from the 16S table to leave only bacteria. In the end, 11,810 unique bacterial sequences and 32,837 unique fungal sequences remained. Taxonomy was assigned using the SILVA reference database v138.1 for prokaryotes [26,27] and the UNITE database v8.2 for fungi [28], and species-level identification was done only in the case of 100% sequence identity. Microbial community data was organized with phyloseq v.1.34.0 for downstream analysis [29].

All statistical analyses were performed in R software v4.0.3 (R Core Team 2020). Soil physicochemical data were scaled to have unit variance and ordinated with principal components analysis (PCA) using the prcomp function of the stats package (stats::prcomp, R Core Team 2020). Effects of neighboring shade tree species on soil properties, microbial alpha diversity metrics, and cacao yields were tested with analysis of variance (ANOVA) followed by Tukey's HSD test. Microbial alpha diversity was investigated by calculating richness, the Shannon index, and the Pielou index with vegan v.2.5-7 [30]. Bacterial and fungal ASV tables were log-transformed and ordinated using principal coordinates analysis (PCoA) based on Bray–Curtis distances (phyloseq::ordinate). Subsequently, significant differences among cacao rhizosphere microbial communities adjacent to different shade tree species were tested with analysis of similarity (ANOSIM), which is robust to unequal group size (vegan::anosim). The spatial variability of microbial communities (distance-decay relationship) was tested with multiple matrix regression analysis using the method described by [31] with 999 permutations.

Stepwise regression was used to identify the soil physicochemical parameters most important for microbial communities and the soil and microbial variables most important for cacao yields. Prior to conducting stepwise regression, variables were inspected for correlation using stats::cor and highly correlated variables were removed to avoid collinearity. A single outlier observation of soil P was removed as it was considered to be above a realistic range, and highly skewed variables were log-transformed to achieve a more normal distribution. Predictor variables included noncollinear soil physicochemical variables, fungal and bacterial alpha diversity metrics, and the first two eigenvectors of bacterial and fungal communities (Table 1).

**Table 1.** Microbial parameters included in stepwise regression.

| Term | Category | Description |
|---|---|---|
| PC1B | Community composition | First eigenvector of bacterial community |
| PC2B | Community composition | Second eigenvector of bacterial community |
| PC1F | Community composition | First eigenvector of fungal community |
| PC2F | Community composition | Second eigenvector of fungal community |
| RichnessB | Diversity | Number of bacterial ASVs |
| ShannonB | Diversity | Shannon index for bacteria (measure of richness and evenness) |
| RichnessF | Diversity | Number of fungal ASVs |
| ShannonF | Diversity | Shannon index for fungi (measure of richness and evenness) |

The train function of caret v6.0.86 [32] was used to perform stepwise regression using the leapSeq method with 10-fold cross-validation. Models with and without microbial predictors were compared using stats::AIC().

Because stepwise regression revealed that PC1B, PC2B, and PC2F were important predictor variables for cacao yields, we explored the taxonomic identity and relevant functions of the 100 ASVs with the highest loadings for each eigenvector. The microeco

package [33] was used to implement functional prediction for prokaryotic sequences with the FAPROTAX database [34]. The potential ecological roles of fungi were investigated by assigning taxa to their primary functional guilds using the FUNGuild [35] and Fungal-Traits [36] databases.

### 3. Results

*3.1. Shade Tree Presence Increased pH and P and Altered N Speciation in the Cacao Rhizosphere*

　　Ordination with PCA showed slight clustering of the samples with adjacent shade trees (Diverse-Cacao) separately from samples with only adjacent cacao trees (Cacao-Cacao) (Figure 2). Separation occurred primarily along the first eigenvector, which accounted for 25% of variation among samples.

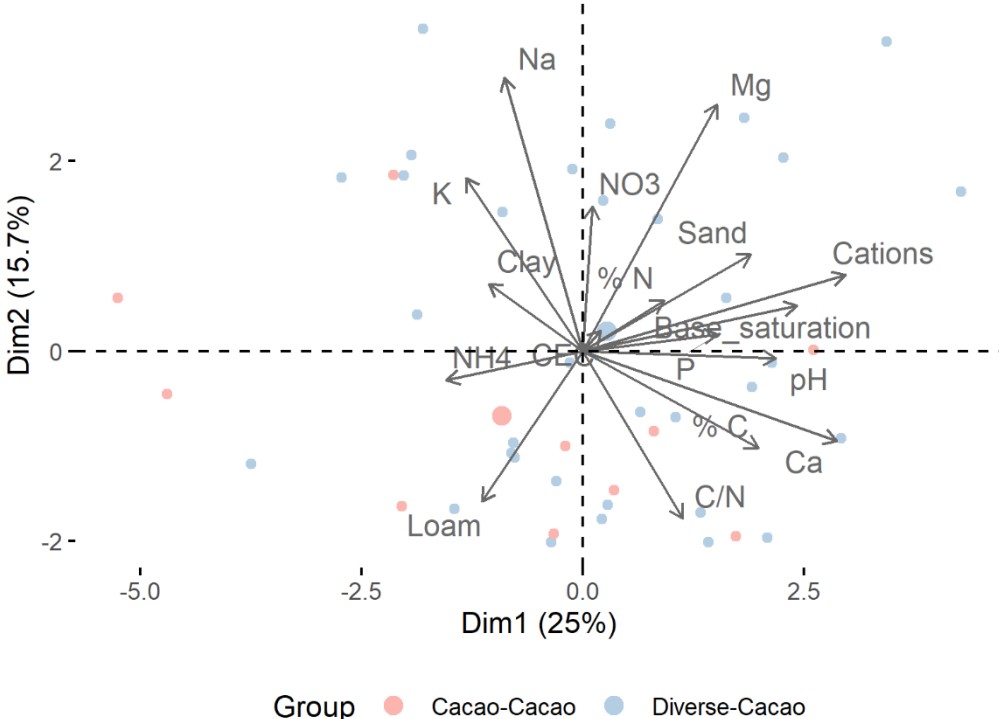

**Figure 2.** Principal components analysis (PCA) ordination of soil physicochemical properties. Soil sampled from cacao with adjacent shade trees (Diverse-Cacao treatment) clustered separately from cacao without adjacent shade trees in the same plot (Cacao-Cacao treatment). Points represent individual soil samples, and larger points represent group means. CEC: cation exchange capacity, C/N: carbon/nitrogen ratio.

　　Further comparison of soil properties between the two treatments showed that Diverse-Cacao samples had higher pH ($p < 0.001$), higher P and $NO_3$-N (both $p < 0.05$), and slightly lower $NH_4$-N ($p < 0.05$) than Cacao-Cacao samples (Figure S1). On average, the presence of shade trees raised pH by 0.3 units (7.05 vs. 6.72), P by almost 70 ppm (97 ppm vs. 28 ppm), and $NO_3$-N by 20 ppm (57 ppm vs. 36 ppm) while decreasing $NH_4$-N by 13 ppm (47 vs. 34 ppm).

　　Despite evidence of the effects of shade tree *presence* on cacao rhizosphere soil properties, shade tree *identity* did not affect $NO_3$-N, $NH_4$-N, and P concentrations (ANOVA, $p > 0.05$). Only pH was affected by shade tree species ($p < 0.01$), and Tukey's HSD tests revealed that pH was higher in the rhizosphere of cacao trees with neighboring red teak, sengon, or banana trees than that in cacao with neighboring cacao.

*3.2. Shade Trees Did Not Affect Cacao Rhizosphere Microbial Diversity, Community Composition, or Spatial Distribution*

Alpha diversity of bacterial and fungal communities in the cacao rhizosphere was not affected by the presence of shade trees, regardless of the diversity index used (Figure S2). Bacterial richness ranged from 36 to 521 unique ASVs per sample, much lower than the observed fungal richness range of 521–1423 unique ASVs.

Community composition of cacao rhizosphere bacteria and fungi was similarly un-affected by the presence or identity of neighboring shade trees (Figure 3). Analysis of similarity (ANOSIM) showed that differences among microbial communities were not due to adjacent shade tree species (both bacteria and fungi $p > 0.05$).

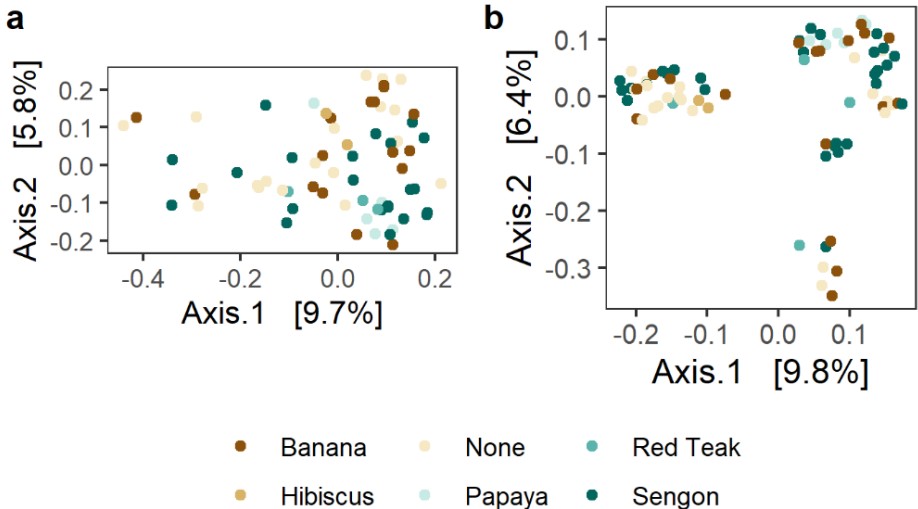

**Figure 3.** Principal coordinates analysis (PCoA) based on Bray–Curtis distance of bacterial and fungal communities. No effect of shade tree presence or species identity was observed for either (**a**) bacteria or (**b**) fungi (both ANOSIM $p > 0.05$). Points represent cacao rhizosphere microbial communities, and colors indicate the species of the nearest shade tree.

Microbial distance-decay relationships, which reflect how between-sample (beta) diversity changes with geographic distance, did not show different patterns between Diverse-Cacao and Cacao-Cacao samples. Multiple matrix regression revealed significant distance-decay relationships, where bacterial communities more geographically distant from one another were more dissimilar, in both sets of samples. The decrease in between-sample similarity with distance was slightly more pronounced in the Diverse-Cacao samples ($R^2 = 0.254$, $p < 0.001$) than that in the Cacao-Cacao samples ($R^2 = 0.084$, $p < 0.01$). In contrast, fungal distance-decay relationships were slightly stronger in Cacao-Cacao samples (Diverse-Cacao: $R^2 = 0.108$, $p < 0.001$; Cacao-Cacao: $R^2 = 0.247$, $p < 0.001$).

*3.3. Cacao Yields Were Not Directly Affected by Shade Trees*

Cacao yields of dry beans, healthy pods, and infected pods were unaffected by the presence or identity of adjacent shade trees (Figure 4). Although the highest yields were measured in Cacao-Cacao trees, yields showed a high degree of variability, and differences were not significant (all $p > 0.05$).

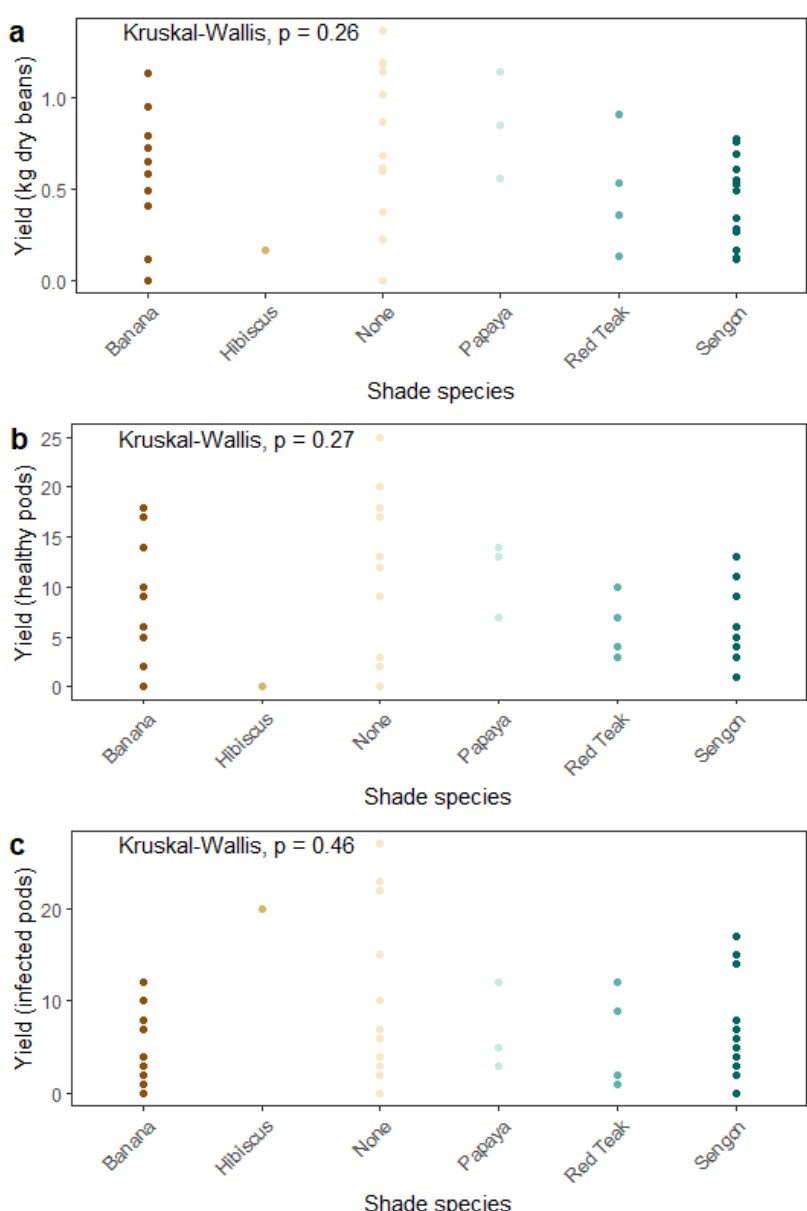

**Figure 4.** Effects of shade tree presence and identity on cacao yield metrics. (**a**) Dry beans, (**b**) healthy pods, and (**c**) infected pods did not differ significantly according to presence or species of the nearest shade tree. "None" indicates yields of cacao trees without neighboring shade trees (Cacao-Cacao treatment).

*3.4. Microbial Variation Was Driven by Soil Properties*

Stepwise regression identified key soil properties with the greatest effect on microbial diversity and community composition (Figure 5). Bacterial diversity responded strongly to K, %N, and $NH_4$-N, while bacterial community composition was driven by P and secondarily by %N for the first principal component, PC1B, and Ca for the second principal component, PC2B. Fungal diversity was affected most strongly by %N, and fungal community composition was driven by soil texture and Na (first principal component, PC1F) and Mg and $NO_3$-N (second principal component, PC2F).

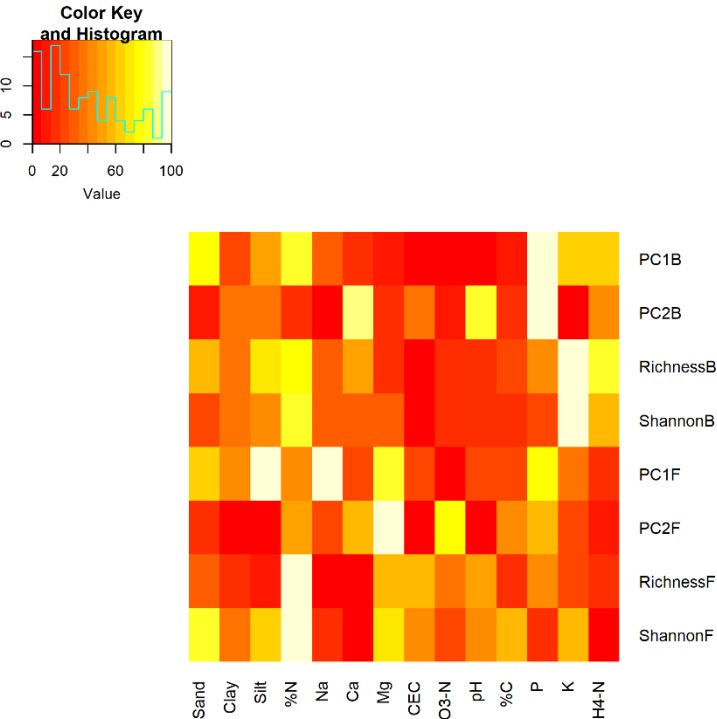

**Figure 5.** Heatmap showing variable importance according to stepwise regression analysis of soil physicochemical parameters affecting microbial diversity and community composition. Brighter colors indicate more important variables in the final regression model, with colors mapped to a variable importance scale of 0–100 (see histogram in upper left). Stepwise regression was implemented with 10-fold cross-validation. CEC: cation exchange capacity; PC1: first eigenvector of community composition; PC2: second eigenvector of community composition; B: bacteria; F: fungi.

### 3.5. Soil Microbial Communities Strongly Influenced Cacao Yields

Stepwise regression was also used to identify soil abiotic and biotic variables predictive of variation in cacao yields. The regression was run twice, first with a set of predictor variables consisting only of soil physicochemical properties (Figure 6a) and then again with microbial diversity and composition added (Figure 6b), to compare the relative influence of soil abiotic and biotic components on yields. When only soil physicochemical variables were considered, soil texture and $NO_3$-N had the greatest influence on dry beans, healthy pods, and infected pods (Figure 6a). Including principal components of bacterial and fungal community composition and alpha diversity metrics showed that microbial variables were also surprisingly strong predictors of cacao yields, in some case stronger than soil physicochemical properties (Figure 6b). Fungal diversity as measured by the Shannon index had the greatest influence on dry bean and healthy pod yields, while bacterial diversity and composition had the greatest influence on infected pod yields (Figure 6b). Fungal community composition was also an important variable for models of both healthy and infected pod yields (PC2F, Figure 6b). All models were improved by the inclusion of microbial variables (Table 2).

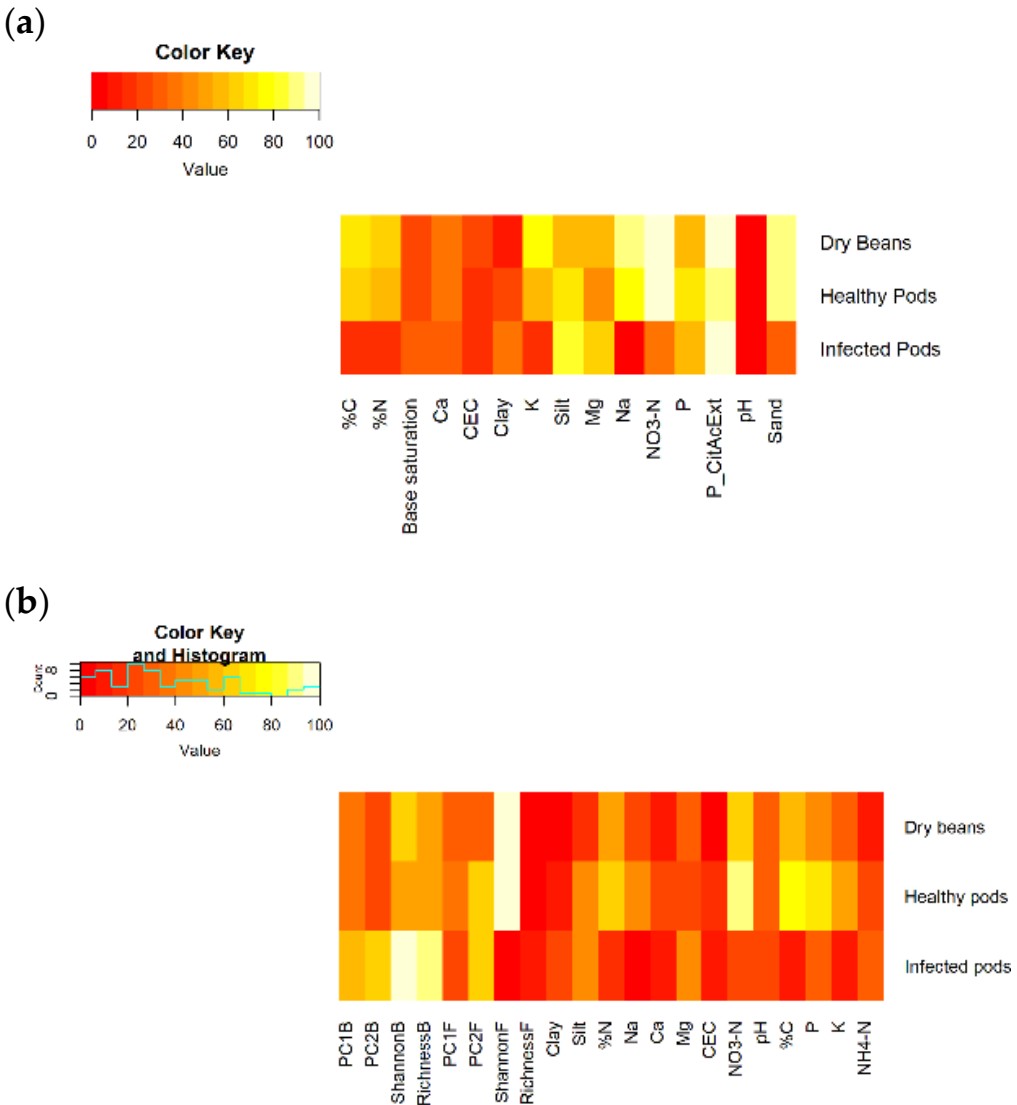

**Figure 6.** Heatmaps showing variable importance in stepwise regression models for cacao yield metrics using (**a**) only soil physicochemical properties as predictor variables and (**b**) both microbial and soil physicochemical parameters as predictor variables. Brighter colors indicate more important variables in the final model after 10-fold cross-validation, with colors mapped to variable importance on a scale of 0–100 (see histogram in upper left). CEC: cation exchange capacity; PC1: first eigenvector of community composition; PC2: second eigenvector of community composition; B: bacteria; F: fungi.

**Table 2.** Model parameters for stepwise regression analysis. Stepwise regression showed that models using both microbial and soil physicochemical predictor variables had lower AIC values, indicating a better fit, than models with only soil physicochemical variables.

| Response Variable | Soil Physicochemical Predictors | Soil and Microbial Predictors |
|---|---|---|
| Dry beans | df: 3, AIC: 39.70 | df: 6, AIC: 20.08 |
| Healthy pods | df: 3, AIC: 302.89 | df: 4, AIC: 246.94 |
| Infected pods | df: 3, AIC: 314.29 | df: 3, AIC: 263.07 |

To further explore the importance of microbial variables, we tested whether these variables were positively or negatively correlated with any yield metrics. Fungal diversity (Shannon index) was significantly positively correlated with both healthy pods (Kendall $\tau = 0.26$, $p = 0.020$) and dry bean yields ($\tau = 0.28$, $p = 0.012$). However, despite their

importance in the stepwise regression model (Figure 6b), bacterial diversity (Shannon index) and community composition (PC1B, PC2B) were not correlated with infected pod yields (Shannon $\tau = 0.065$, $p > 0.05$; PC1B $\tau = 0.032$, $p > 0.05$; PC2B $\tau = -0.043$, $p > 0.05$). PC2F was not correlated with either healthy or infected pod yields (healthy $\tau = 0.098$, $p > 0.05$; infected $\tau = 0.026$, $p > 0.05$).

We then characterized the taxonomy and function of the 100 ASVs with the greatest contribution to PC1B, PC2B, and PC2F due to the importance of those variables for healthy and infected pod yields. The Proteobacteria were the most represented phylum in both PC1B (Figure S3a) and PC2B (Figure S3b). The phyla Verrucomicrobiota and Acidobacteriota were more abundant in PC1B than in PC2B, while Plantomycetota and Bacteroidota were more abundant in PC2B. In PC2F, the Ascomycota were the most represented phylum, followed by the Basidiomycota, and the Sordariomycetes were the most-represented class (Figure S3c).

Functional prediction of PC1B with the FAPROTAX database showed that the most abundant functions were chemoheterotrophy (29% of ASVs) and aerobic chemoheterotrophy (28%), with 0% plant pathogens. Similarly, 25% of ASVs in PC2B were capable of chemoheterotrophy (with the same percentage capable of aerobic chemoheterotrophy), and no plant pathogens were present. Classification of primary lifestyles of fungi belonging to PC2F using the FungalTraits database revealed that the highest percentages were plant pathogens (17%), followed by litter and wood saprotrophs (11% each).

## 4. Discussion

We found that early in the establishment of this diversified CAFS, shade trees improved important soil properties without decreasing cacao yields, and that microbial diversity and community composition had a surprisingly strong influence on cacao productivity. In partial support of our second hypothesis, shade tree presence (but not species identity) improved yield-relevant physicochemical properties in cacao rhizosphere soil (Figure 7).

Shade tree presence was associated with higher $NO_3$-N, P, and pH in the cacao rhizosphere (Figure S1), and stepwise regression showed that $NO_3$-N was the most important soil physicochemical variable in predicting dry bean and healthy pod yields (Figure 6a). P and pH can limit yields in tropical soils, so it is worth highlighting the beneficial impacts of shade trees on these properties even though they did not show up as highly significant predictors of yield in stepwise regression. That increased soil nutrients near shade trees did not translate to higher yields in this study may be a consequence of measuring young cacao trees not yet bearing full harvests within the restricted time horizon of a single harvest year. Soil macronutrient increases due to shade trees do not necessarily result in increased foliar nutrient concentrations in the same growing season [37]. The effects of shade trees on soil and microbial parameters are likely to become more pronounced over time as leaf litter accumulates and decomposes, so further improvements in soil properties may occur in the long term.

The positive impacts of shade trees on soil fertility observed in the cacao rhizosphere here are consistent with some reported improvements in bulk soil or shade-tree-associated soil in the literature. Higher soil N directly under shade trees was found in [16], though there were no benefits to soil fertility parameters of increasing shade tree cover at the plot scale. However, the increases in P and $NO_3$-N reported here differ from a comparison of three contrasting shade trees in Ghana, where P was decreased under all three species, and total N was not affected by presence or identity of shade trees [9]. Resource availability may help determine the nature of interactions between cacao and shade species, favoring commensal or beneficial relationships when resources are plentiful and triggering competition when soil nutrients are scarce. We emphasize that CAFS represent a highly diverse range of agroecosystems, and the specific impacts of shade tree integration on soil properties likely depend greatly on geographic and agronomic contexts.

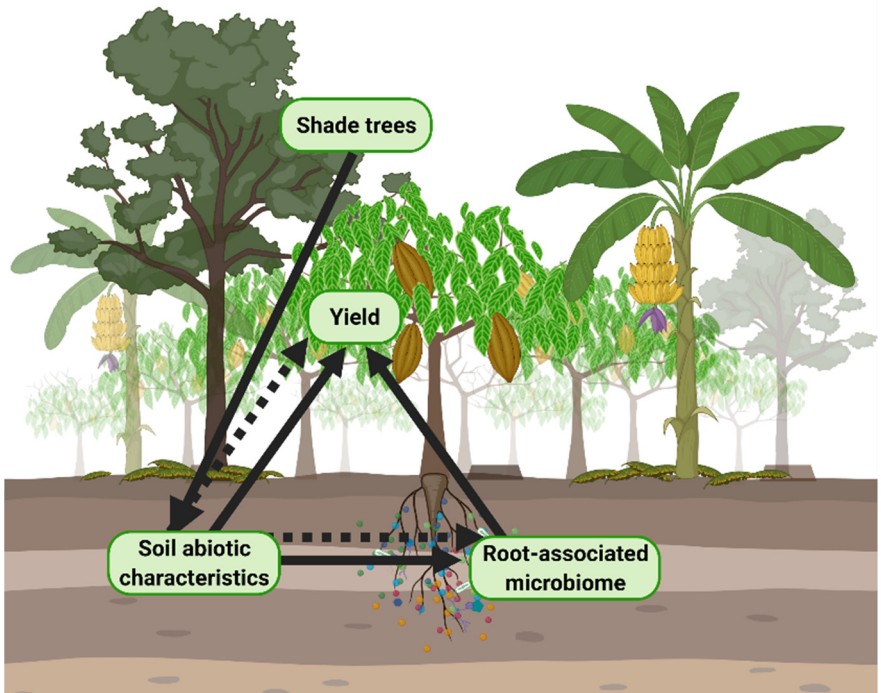

**Figure 7.** Relationships among shade trees, soil physicochemical properties, root-associated microbial communities, and cacao yields in this study. Shade trees had direct impacts only on soil physicochemical properties, specifically increasing pH and P while altering N speciation. However, shade trees also affected yield and microbial communities indirectly via their influence on soil properties, as soil N impacted yield, P affected fungal richness, and pH affected bacterial and fungal community composition. Other soil properties not influenced by presence or identity of shade trees also impacted yield and microbial communities. Microbial diversity and community composition were both important variables in predicting cacao yields.

Contrary to our first hypothesis, diversification did not incur a yield penalty: cacao yields were not directly affected by either presence or identity of adjacent shade trees (Figure 7). Large-canopied trees such as sengon intercept much more light than smaller-canopied species such as hibiscus and banana, and increasing shade cover has been found to be associated with decreasing cacao growth and yields [16]. This trial is still newly established, however, and effects of adjacent shade species on cacao growth that are driven by canopy size, competition for nutrients, and other factors will likely be observed only at later stages of maturity. Whether increased competition for light and other resources by maturing shade trees will outweigh their previously mentioned compounding positive effects on soil fertility over time remains an open question. Measuring seven to twelve years after plot establishment, for example, another study of a diversified CAFS in South Sulawesi observed differences in cacao yields under commonly planted shade tree species (e.g., gliricidia, rambutan, langsat, durian, jackfruit, jabon, guava, mango, petai, coconut, gmelina) [14]. However, no effect of shade tree species on healthy pods or total pod count was found in a comparison of leguminous and timber shade trees intercropped with cacao in the initial 10–11 years after establishment [38]. Similarly, cacao seedling growth did not differ in soils isolated from under different shade species in traditional cabruca CAFS in Brazil [39], and shade tree diversity did not affect cacao yields across a diversification gradient in Southeast Sulawesi [13]. Cacao yields are highly variable and influenced by numerous factors in addition to shade tree presence and identity, such as cacao genetics, tree maturity, soil fertility and resource availability, climate, shade tree density and proximity to cacao trees, and shade canopy size, so it is not surprising that results are context-dependent. In light of the large genetic contribution to yield variation in cacao [40], comparisons between studies may be of limited usefulness unless based on the same clones.

This study is the first to our knowledge to assess impacts of shade trees on cacao rhizosphere microbial communities at tree and plot scales. Contrary to our hypothesis that shade tree presence and species identity would directly affect microbial alpha diversity, community composition, and distribution within the plot (Figure 1), only distance-decay relationships were even moderately affected by shade tree presence, and the direction of the effect differed between bacteria and fungi. Bacterial and fungal communities are certainly expected to respond to inputs of shade-tree-derived carbon sources such as leaf litter, senescing fine roots, and root exudates, but in this study at least, the zone of influence of shade trees did not extend to the cacao rhizosphere. Relative immaturity of these shade trees, as noted with regard to yield, may account for limited impacts of shade presence and species.

Nonetheless, even these young shade trees affected microbial communities indirectly via soil properties, with downstream impacts on yield (Figure 7). Shade tree presence increased P (Figure S1), the primary predictor of PC1B and PC2B (Figure 5). Presence of shade trees also altered $NO_3$-N and $NH_4$-N (Figure S1), which impacted bacterial richness and PC2F (Figure 5), relevant variables for healthy and infected pod yields (Figure 6b). Shade tree presence and species identity both affected soil pH (Figure S1), which was a strong predictor of PC2B (Figure 5), which in turn was an important predictor of infected pod yields (Figure 6b). Given that functional prediction of PC2B with FAPROTAX did not reveal any known plant pathogens, it may be that PC2B contains taxa capable of biocontrol or that pathogen suppression or facilitation are emergent properties of the ecological interactions among the members of PC2B.

This study highlights surprising and strong impacts of bacterial and fungal diversity on cacao yields and pod infection rates (Figure 6b). Whether microbial diversity per se is desirable in agricultural contexts is a recurring topic in discussions of how best to promote soil health; although these results do not clarify the mechanisms involved, they do provide support for managing CAFS to promote rhizosphere microbial diversity. Beyond diversity, specific taxa are likely also important for yields, although even taxonomic assignment, functional prediction, and functional guild assignment were not able to provide much insight into the mechanisms involved. The principal components of microbial community composition shown to affect yields in stepwise regression are worthy of future investigation to identify cacao yield-promoting taxa. Interestingly, PC2F contained both taxa well-established as beneficial for cacao (e.g., *Trichoderma*) and taxa that contain plant pathogens (*Magnaporthe, Fusarium*) (Figure S3c), with 17% of ASVs classified as plant pathogens in the FungalTraits database. Much remains to be learned about the diverse ecological roles of these fungi in CAFS, however.

In sum, we found that shade trees enhanced soil fertility in the cacao rhizosphere without yield penalties early in establishment, and that shade tree presence may have indirect, rhizosphere-mediated benefits for cacao yields in the long term. At this early stage, minimal variation was observed among diverse shade species in their impacts on rhizosphere soil abiotic and biotic properties. Choice of shade trees in this specific agroecological context may thus be based on factors other than cacao yields, including established benefits such as habitat for migratory birds and alternative sources of income for farmers [41]. This observation is particularly important because many farmers choose CAFS over monoculture primarily for the alternative income sources represented by timber or fruit-bearing shade trees, and economic sustainability ensures that environmental benefits will be realized over the long term. We further highlight a surprisingly strong influence of the cacao rhizosphere microbiome on pod and bean yields and emphasize the particular importance of microbial diversity. This study thus adds to the numerous known reasons to integrate shade trees in cacao production systems by showing specific beneficial impacts of shade tree presence on the cacao rhizosphere.

**Supplementary Materials:** The following supporting information can be downloaded at: https://www.mdpi.com/article/10.3390/agronomy12010195/s1, Figure S1: Effects of adjacent shade trees on cacao rhizosphere soil physicochemical parameters; Figure S2: Effects of shade tree presence on microbial alpha diversity; Figure S3: Phylogenetic heat trees of yield-relevant microbial taxa; Table S1: Plant species included in trial; Table S2: Soil amendments at planting; Table S3: Soil amendments over time.

**Author Contributions:** Conceptualization, J.E.S., A.F. and S.J.F.; methodology, J.E.S., A.F., N.I.I., T.M.C. and S.J.F.; formal analysis, J.E.S., T.M.C. and A.F.; investigation, H.H.; data curation, H.H. and S.J.F.; writing—original draft preparation, J.E.S., N.I.I. and S.J.F.; writing—review and editing, J.E.S., A.F. and S.J.F.; visualization, J.E.S.; supervision, J.E.S. and S.J.F.; project administration, A.F. All authors have read and agreed to the published version of the manuscript.

**Funding:** This research received no external funding.

**Institutional Review Board Statement:** Not applicable.

**Informed Consent Statement:** Not applicable.

**Data Availability Statement:** The data presented in this study are openly available in the NCBI SRA under BioProject ID PRJNA786222. Code for the analyses is available at https://github.com/jes12010/microbiome-diversification-variability (accessed on 20 December 2021).

**Acknowledgments:** The authors would like to thank the entire Farm Income Diversification Team of the Tarengge Mars Cocoa Research Centre for their field sampling assistance and Thomas Fungenzi for his technical advice. The authors would also like to thank the ICCRI laboratory for sharing a translation of their analytical methods.

**Conflicts of Interest:** All authors are employed by Mars Inc.

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
