# Peer review of "Impacts of Shade Trees on the Adjacent Cacao Rhizosphere in a Young Diversified Agroforestry System"

_agronomy, doi:10.3390/agronomy12010195_

Round 1

Reviewer 1 Report

General comments

The effects of plant diversity on the functioning of agroforestry systems are most often analyzed at the level of the above-ground compartment. Working on cacao agroforestry systems, here referred to as CAFs, this manuscript aims at studying the effects of shade trees, i.e. trees above cacao trees, on the rhizosphere surrounding cacao trees. The topic is clearly addressed and the contribution of this paper is valuable bringing interesting results on young cacao agroforestry systems.

The writing is clear, with a well-developed introduction and sound discussion. However, I found that Materials and Methods and Results should be more instructive about the variables analyzed and discussed. Illustrations are well presented. Please check references to supplementary figures that are not all mentioned in the text.

Specific comments

Title: The fact that the experiment is carried out on young CAFS, including both young cacao trees and shade trees, is important in terms of interpretation of results and possibility to generalize (see Discussion, Li501, Li530). This fact should be mentioned in the title.

Figure 1: “soil” has a general meaning and may include abiotic and biotic characteristics. In the figure itself I would complete by “abiotic characteristics” as written in Li85.

Li112-113: “Note” to remove as well as italics and keep “The cultivation… agrochemicals” as a sentence.

Li119: how many cacao trees among these 703 plants?

Li120: “fruits, and timber” as shade trees also?

Li130: are they own-rooted clones or with a rootstock?

Li157: add “CPB” before Conopomorpha

Li172: an effect of cacao clone on either diverse-cacao or caca-cacao interactions?

Li238: as above, do not put a “note”. It can be a sentence in itself.

Li244: same percentage for the other clone?

Li254: add reference [24] for R core team.

Li286: log transformation?

Li290: I recommend to add a table with all acronyms at the end of Materials and Methods. This will facilitate the understanding of results. Here PC1B etc. appear and it is important to know what you mean by this. The same for “Shannon index” that stands for diversity (Li412).

Legend of Figure 2: add that large symbols are means for each group.

Legend of Figure 3, to complete, add (a) and (b), and remove the last sentence which is not relative to this figure.

Li355: add “by” before “shade trees”.

Figure 5: all acronyms on the right of the large heatmap should have been introduced before (see before table needed).

Figure 6: heatmaps to enlarge.

Figure 6 legend: Li406: do you make a difference between “physicochemical properties” and “soil parameters”. If not, please use always the same terms to specify the same thing.

Li457 (end of paragraph): as written we understand that the system will improve with age including in terms of yield. This is only speculative and should also include the fact that light will decrease and competition among root systems will increase with age. This statement is tempered by what you write in Li476. I suggest to combine these two sentences to make a clear general statement on this. As it is it seems contradictory.

Li483 and following: you mention several studies, but it is likely that cacao genetics plays an important role and comparison between your study on 2 clones and other studies, some of them likely on genetic crosses, should be considered with caution. You may keep these comparisons but mentioning this aspect.

Li653: reference to complete. Is there a web site?

Author Response

Specific comments

Title: The fact that the experiment is carried out on young CAFS, including both young cacao trees and shade trees, is important in terms of interpretation of results and possibility to generalize (see Discussion, Li501, Li530). This fact should be mentioned in the title.

Thank you for the insight. We agree and have added “young” to the title.

Figure 1: “soil” has a general meaning and may include abiotic and biotic characteristics. In the figure itself I would complete by “abiotic characteristics” as written in Li85.

We agree with this point and have revised both Figure 1 and Figure 7 accordingly.

Li112-113: “Note” to remove as well as italics and keep “The cultivation… agrochemicals” as a sentence.

Revised as suggested.

Li119: how many cacao trees among these 703 plants?

There were 253 cocoa trees in total, and this has been added to line 118 as well as line 130.

Li120: “fruits, and timber” as shade trees also?

While fruits and timber trees will indeed provide shade once they develop to maturity, at the time of this study, these species were not providing much shade to cocoa, as their canopies were either below the cocoa tree canopy, or their canopies were very minimal (i.e. teak canopy). The timber tree architecture is tall and slim with a sparse canopy when young. Fruits and timber trees are planted as 'multi-purpose trees' that provide both fruits, shade as well as other unintended benefits such as support to biodiversity. At the time of this study, shade to cocoa was provided by the fast-growing leguminous tree species 'Sengon' (Falcataria moluccana), which was planted specifically as a shade tree and had large developed canopies at the time of the study.

Li130: are they own-rooted clones or with a rootstock?

The majority of the cocoa trees (244) are scion cuttings of clone MCC02, top-grated on seed-grown rootstocks of clone BB01 seeds. However, some clones (9) have BB01 propagated on top of BB01 seeds. This is because MCC02 is a highly self-incompatible clone and a source clone for cross compatible pollen is thus required within the field. Clone BB01 has been found as the most highly cross-compatible clone with MCC02 available. The information about number of trees per clone and grafting has been added to line 130.

Li157: add “CPB” before Conopomorpha

Revised as suggested.

Li172: an effect of cacao clone on either diverse-cacao or caca-cacao interactions?

While there were nine BB01 clones in the diversification plot, we did not sample the soil surrounding BB01 cocoa trees. This is now reflected in the text (line 165). We were constrained by sampling costs so opted only to sample the one clone, especially considering the low number of replicates of clone BB01. However, we do appreciate that this is a very interesting concept and should be followed up in future studies.

Li238: as above, do not put a “note”. It can be a sentence in itself.

Revised as suggested.

Li244: same percentage for the other clone?

Because only this clone was sampled, as described in our response above and added to line 165, only this percentage is relevant.

Li254: add reference [24] for R core team.

Reference added.

Li286: log transformation?

Yes, revised to read “log-transformed”.

Li290: I recommend to add a table with all acronyms at the end of Materials and Methods. This will facilitate the understanding of results. Here PC1B etc. appear and it is important to know what you mean by this. The same for “Shannon index” that stands for diversity (Li412).

Thank you for the suggestion. We have added a new table (Table 1) at the end of that section clarifying all of the microbial parameters used in stepwise regression and their interpretation. In addition, all of the terms were already explained in the methods and the figure legends.

Legend of Figure 2: add that large symbols are means for each group.

Revised as suggested: “Points represent individual soil samples and larger points represent group means” (line 309).

Legend of Figure 3, to complete, add (a) and (b), and remove the last sentence which is not relative to this figure.

Revised as suggested.

Li355: add “by” before “shade trees”.

Thank you for catching this error. Revised as suggested.

Figure 5: all acronyms on the right of the large heatmap should have been introduced before (see before table needed).

These have now been introduced in the new Table 1.

Figure 6: heatmaps to enlarge.

We have enlarged figures 6a and 6b and placed them in a single column to accommodate the larger size. We note that all figure formatting is ultimately decided by the journal and understand that our figures may be reformatted during publication.

Figure 6 legend: Li406: do you make a difference between “physicochemical properties” and “soil parameters”. If not, please use always the same terms to specify the same thing.

Thank you for catching this oversight. We have revised this line to specify “soil physicochemical parameters”.

Li457 (end of paragraph): as written we understand that the system will improve with age including in terms of yield. This is only speculative and should also include the fact that light will decrease and competition among root systems will increase with age. This statement is tempered by what you write in Li476. I suggest to combine these two sentences to make a clear general statement on this. As it is it seems contradictory.

Thank you for highlighting this important point. We have rewritten line 457 to clarify that our focus in that sentence was long-term improvements in soil properties, which might then favorably impact yield. To address your second suggestion, we have added a sentence (lines 474-476 in revised version) to clarify that it is still unclear whether long-term improvements in soil fertility would balance out the effects of increased competition for light and other resources: “Whether increased competition for light and other resources by maturing shade trees will outweigh their previously mentioned compounding positive effects on soil fertility over time remains an open question.”

Li483 and following: you mention several studies, but it is likely that cacao genetics plays an important role and comparison between your study on 2 clones and other studies, some of them likely on genetic crosses, should be considered with caution. You may keep these comparisons but mentioning this aspect.

Thank you for this point. We agree that cacao genetics are a very important source of yield variability, although we do not compare two clones in this study. We have added cacao genetics to the list of factors that may contribute to context-dependent results of shade tree studies in line 483 and have added another sentence after that to further highlight their importance: “In light of the large genetic contribution to yield variation in cacao [41], comparisons between studies may be of limited usefulness unless based on the same clones.”

Li653: reference to complete. Is there a web site?

This is a dissertation and was not published as a single paper, so the reference format differs from the other references. The URL has been added.

Reviewer 2 Report

I was happy to read this manuscript presenting very interesting research. I like that this research was relatively simple, but very well designed and all the data rigorously evaluated. The results form this research should be of the interest of wider scientific community focusing on cocoa agroforestry. The results are presented in efficient and clear manner and I found them really interesting and significant for further research. It clearly confirms the importance of diversification of cocoa growing with various trees.

I have just some minor comments.

Introduction
- The objectives are clearly stated
- I guess that there is no need to show Figure 1. - it is quite general and can be easily explained only in the text, probably Fig. 7 is enough
Materials and methods
- Very clear description of a study site and experiment 
- Please add Latin names of the plants throughout the whole ms - e.g. gliricidia - G. sepium I guess, pineapple, chilli, senton??? etc….
- It would be good to identify (at least as a supplementary table) the list of crops, fruit and timber trees that were planted in the experimental plot. Also the figure of the experimental plot would be helpful to understand it.
- Soil sampling - why you choose only 48 samples for soil chemical analysis out of 80?
Results
- Clearly end efficiently presented 
- Could you please increase the size of figure 6 - it is difficult to read
- I am not sure if I read well figure 5 and 6. Brighter colors means more yellow or white? And it means greater influence?
Discussion
- Please bring figure 7 closer to the reference in the text
- Very well written
- How distant were the trees from the cocoa plants? Did you ale checked that influence?

Author Response

Introduction
- I guess that there is no need to show Figure 1. - it is quite general and can be easily explained only in the text, probably Fig. 7 is enough

Thank you for this suggestion. In light of comments from another reviewer, we have revised Figure 1 slightly but have kept it in the manuscript. We will ultimately leave it up to the editor to decide if Figure 1 should be eliminated.

Materials and methods
- Very clear description of a study site and experiment 
- Please add Latin names of the plants throughout the whole ms - e.g. gliricidia - G. sepium I guess, pineapple, chilli, senton??? etc….

Thank you for the suggestion. We have added this information to the new supplementary Table S1.
- It would be good to identify (at least as a supplementary table) the list of crops, fruit and timber trees that were planted in the experimental plot. Also the figure of the experimental plot would be helpful to understand it.

We have added a new supplementary Table S1 with a full description of plant species common names, scientific names, and numbers of plants.

- Soil sampling - why you choose only 48 samples for soil chemical analysis out of 80?

Due to the cost of soil physicochemical analysis, we were only able to analyze a subset of the samples. We have added that information to line 170.

Results
- Clearly end efficiently presented 

Thank you.
- Could you please increase the size of figure 6 - it is difficult to read

Thank you for pointing this out. We have increased the size of figure 6.
- I am not sure if I read well figure 5 and 6. Brighter colors means more yellow or white? And it means greater influence?

Yes, as described in the legend, “Brighter colors indicate more important variables in the final regression model.” The histogram in the upper left-hand corner of each figure maps variable importance to colors. We have expanded the description in the legend to include “with colors mapped to a variable importance scale of 0-100 (see histogram in upper left).”

Discussion
- Please bring figure 7 closer to the reference in the text

Thank you for this suggestion. We have moved Figure 7 to the beginning of the discussion, where it is first mentioned in the text.
- Very well written

Thank you.
- How distant were the trees from the cocoa plants? Did you ale checked that influence?

In this analysis, we considered only shade trees that were directly adjacent to the cacao trees, so there were no other trees between them and they were close. We did take GPS coordinates of each shade tree and cacao tree and ran an initial analysis to determine whether effects of shade trees were significant if we considered only trees within a certain distance of the cacao tree. Because this analysis did not show an effect of shade tree proximity (i.e. shade tree presence effects were still not significant) and because GPS data is limited in accuracy at a spatial resolution (+/- 3m) that may include differences in distances among shade trees, we decided not to include that analysis in the paper.